# Activity and Anti-Aflatoxigenic Effect of Indigenously Characterized Probiotic Lactobacilli against *Aspergillus flavus*—A Common Poultry Feed Contaminant

**DOI:** 10.3390/ani9040166

**Published:** 2019-04-15

**Authors:** Nimra Azeem, Muhammad Nawaz, Aftab Ahmad Anjum, Shagufta Saeed, Saba Sana, Amina Mustafa, Muhammad Rizwan Yousuf

**Affiliations:** 1Department of Microbiology, University of Veterinary and Animal Sciences, Lahore 54000, Punjab, Pakistan; nimra_azeem2011@hotmail.com (N.A.); aftab.anjum@uvas.edu.pk (A.A.A.); saba.sana@uvas.edu.pk (S.S.); aminamustafa046@gmail.com (A.M.); 2Institute of Biochemistry and Biotechnology, University of Veterinary and Animal Sciences, Lahore 54000, Punjab, Pakistan; shagufta.saeed@uvas.edu.pk; 3Department of Theriogenology, University of Veterinary and Animal Sciences, Lahore 54000, Punjab, Pakistan; mryousuf@uvas.edu.pk

**Keywords:** Aflatoxin B1, *Lactobacillus*, anti-fungal, *Aspergillus flavus*, in vitro, poultry

## Abstract

**Simple Summary:**

Mycotoxicosis in poultry has been seriously damaging the poultry production in Pakistan, resulting in economic losses to the country. The present study may act as a preliminary step for exploring the effect of indigenously characterized potential probiotic lactobacilli on aflatoxin production by *Aspergillus*
*flavus*. The present study explored anti-fungal *Lactobacillus* strains. Further investigations revealed their in vitro aflatoxin binding and anti-aflatoxigenic capabilities. These findings demonstrated *L. gallinarum* PL 149 to be an effective binder of aflatoxin B1 which may be used as a biocontrol agent against *A. flavus* and aflatoxin B1 production. It may be further employed for aflatoxin binding in poultry gut after in vivo evaluations.

**Abstract:**

Aflatoxin contamination in human food and animal feed is a threat to public safety. Aflatoxin B1 (AFB1) can be especially damaging to poultry production and consequently economic development of Pakistan. The present study assessed the in vitro binding of AFB1 by indigenously characterized probiotic lactobacilli. Six isolates (*Lactobacillus gallinarum* PDP 10, *Lactobacillus reuetri* FYP 38, *Lactobacillus fermentum* PDP 24, *Lactobacillus gallinarum* PL 53, *Lactobacillus paracasei* PL 120, and *Lactobacillus gallinarum* PL 149) were tested for activity against toxigenic *Aspergillus flavus* W-7.1 (AFB1 producer) by well diffusion assay. Only three isolates (PL 53, PL 120, and PL 149) had activity against *A. flavus* W-7.1. The ameliorative effect of these probiotic isolates on AFB1 production was determined by co-culturing fungus with lactobacilli for 12 days, followed by aflatoxin quantification by high-performance liquid chromatography. In vitro AFB1 binding capacities of lactobacilli were determined by their incubation with a standard amount of AFB1 in phosphate buffer saline at 37 °C for 2 h. AFB1 binding capacities of isolates ranged from 28–65%. Four isolates (PDP 10, PDP 24, PL 120, and PL 149) also ceased aflatoxin production completely, whereas PL 53 showed 55% reduction in AFB1 production as compared to control. The present study demonstrated *Lactobacillus gallinarum* PL 149 to be an effective candidate AFB1 binding agent against *Aspergillus flavus*. These findings further support the binding ability of lactic acid bacteria for dietary contaminants.

## 1. Introduction

Poultry is one of the major sectors playing a role in the enhanced economic activity of Pakistan but still it faces a lot of problems, including mycotoxicosis. Mycotoxins are toxic secondary metabolites of fungal origin, which can cause various diseases and death in animals and humans. Ergot alkaloids, fumonisins, patulin, aflatoxin, citrinin, trichothecenes, ochratoxin A, and zearalenone are all examples of some different mycotoxins. Aflatoxins, produced by *Aspergillus parasiticus*, *Aspergillus flavus*, and *Aspergillus nomius*, are of great importance because of their biological and biochemical effects on living systems [1]. Aflatoxin-producing molds are globally and can flourish on a variety of food and feed commodities during production, processing, storage, and transportation procedures [1,2,3]. These molds can infect crops, especially in hot and humid conditions, resulting in economic loss and adverse effects on consumers’ health.

Aflatoxin is a potent carcinogen, mutagen, contains hepatotoxic and immunosuppressant effects and inhibit several metabolic systems resulting in liver and kidney damage [1,4]. Aflatoxin and citrinin cause increased fragility of the vascular system and produce hemorrhages in body tissues. Among aflatoxins, aflatoxin B1 is the most potent, and it is categorized among class 1 human carcinogens. Different factors including pH, temperature, water activity, available nutrients, and competitive inhibition by other microorganisms can affect aflatoxin production in feed [3]. Appropriate harvesting and storage conditions of crops and feed play important roles in aflatoxin reduction.

Various methods have been employed for the removal or inactivation of aflatoxins, including physical, biological, and chemical methods. Chemical treatments may include roasting, ammoniation, and other solvent extraction techniques. Many aflatoxin binders, like activated carbon and various mineral clays, are commercially available and act as sequestering agents and tightly bind aflatoxin; the resulting binding complex is then excreted from the animal’s body [5]. These toxin binders can restore the nutritional value of the feed, but these chemical methods are unsafe, unhealthy, and expensive [6]. Toxin removal by microorganisms is a promising and economical method for decontaminating raw materials and food [7]. Numerous investigations have reported the inhibitory effects of microbes including actinomycetes, yeast, mold, and bacteria on mold growth and aflatoxin production [3]. Thus, beneficial microorganisms may serve as an alternative therapy for mycotoxicosis.

Anti-mutagenic lactic acid bacteria can remove mutagens from food by physical means [8]. Toxin binding by bacteria occurs through cell wall components, namely polysaccharides or polypeptides. Many researchers have studied this binding mechanism, but the exact mechanism of binding is still unknown [9].

Researchers are paying more attention towards preventing the absorption of aflatoxins in the gastrointestinal tracts of users by the aid of probiotic bacterial supplements in food and feed [10]. According to the World Health Organization (WHO), probiotics are defined as live microorganisms which when administered in adequate amounts exert healthy effects to host [11]. *Lactobacillus*, *Bifidobacterium*, *Enterococcus*, *Saccharomyces*, and *Bacillus* may serve as probiotics.

Lactobacilli can efficiently remove aflatoxins from contaminated broth. The toxin removal mechanism involves sequestration by binding the toxin to the cell wall instead of metabolic degradation [12]. The present study may act as a preliminary step for studying the effect of indigenously characterized potential probiotic lactobacilli on aflatoxin production by *Aspergillus flavus*, so that lactobacilli can be used as biocontrol agents. The present study also assessed the in vitro AFB1 binding capacity of *Lactobacillus* spp., so that these probiotic strains can be employed as toxin binders in place of chemicals in animal feed and thereby the harmful effects of chemical toxin binders can be avoided.

## 2. Materials and Methods

### 2.1. Identification of Isolates

Previously characterized probiotic lactobacilli (*n* = 6) of poultry and fermented food origin [13] and toxigenic *Aspergillus flavus* W-7.1 were procured from the Department of Microbiology, University of Veterinary and Animal Sciences, Lahore, as listed in Table 1. Lactobacilli were revived using De Man, Rogosa, and Sharpe (MRS) agar and identified as describe previously [14]. Fungal strain was cultured on Sabouraud Dextrose Agar (SDA) medium incubated at 37 °C for 5–6 days. Culture and microscopic characters were observed for identification as described previously [15].

### 2.2. Antifungal Activity of Lactobacilli

Antifungal activity of lactobacilli (*n* = 6) was determined by well diffusion assay as described elsewhere [16]. Briefly, SDA medium seeded with fungal spores (10^7^ spores/mL) was poured into sterile Petri dishes and allowed to solidify. Wells were punctured in the medium which were then sealed with sterile molten agar. Cell free supernatant (100 μL) of each lactobacilli strain was added into the respective wells. After 3–4 days incubation at 28 °C aerobically, the diameter of zones of inhibition (mm) was measured.

### 2.3. Effect of Lactobacilli on Aflatoxin Production

The effect of lactobacilli on aflatoxin production by *Aspergillus flavus* was observed by inoculating 1 mL bacterial suspension (1 McFarland) in yeast extract sucrose broth (YESB) supplemented with a standard amount of fungal spores (10^7^ spores/mL), followed by incubation at 28 °C and 100 rpm for 10 days. YESB media supplemented with known fungal spores and plain YESB media without any inoculation were also incubated as positive and negative controls, respectively. After incubation, medium containing lactobacilli and fungus was filtered through Whatman filter paper no 1 and aflatoxin B1 quantity in filtrate was measured by high-performance liquid chromatography (HPLC) and compared with controls [6]. Aflatoxin B1 was detected by HPLC and quantified using the following formulae:(1)$$\mathrm{Quantity}\;\mathrm{of}\;\mathrm{Aflatoxins}\;\left(\frac{\mathrm{ng}}{\mathrm{mL}}\right)=\frac{\mathrm{peak}\;\mathrm{area}\;\mathrm{of}\;\mathrm{sample}\;}{\mathrm{peak}\;\mathrm{area}\;\mathrm{of}\;\mathrm{standard}}\times 100$$
(2)$$\%\;\mathrm{age}\;\mathrm{reduction}=\frac{1-\left(\mathrm{Peak}\;\mathrm{area}\;\mathrm{of}\;\mathrm{AFB}1\;\mathrm{in}\;\mathrm{treatment}\right)}{\left(\mathrm{Peak}\;\mathrm{area}\;\mathrm{of}\;\mathrm{AFB}1\;\mathrm{in}\;\mathrm{control}\right)}$$

### 2.4. Aflatoxin B1 Extraction

For toxin extraction, a previously established protocol was used with modifications [17]. Briefly, broth culture of *Aspergillus flavus* was autoclaved at 121 °C and 15 psi and then homogenized using homogenizer. Twenty-five grams of homogenate was treated with chloroform (90 mL), methanol (10 mL), NaCl (5 g), and distilled water (10 mL) and incubated at 37 °C with continuous shaking (150–160 rpm) for 30 min. Filtration was carried out using Whatman filter paper #4 and filtrate was concentrated in a water bath at 50 °C. Concentrate was ground to fine powder and reconstituted in 3 mL chloroform volume and stored at 4 °C.

### 2.5. Toxin Binding Assay

Standard aflatoxin B1 solution was prepared by the method described elsewhere [18]. Prepared standard aflatoxin solution was then added to sterile phosphate buffer saline (PBS) containing lactobacilli culture (1 McFarland). After 2 h of incubation, cells with bound toxin were separated by centrifugation at 10,000 rpm for 5 min and unbound aflatoxin in supernatant was quantified by HPLC.

### 2.6. High Performance Liquid Chromatography (HPLC)

Aflatoxins were quantified by Agilent HPLC system, 1100 series (Agilent, Santa Clara, CA, USA) as described previously [19]. A mixture of acetonitrile, water, and methanol was used as mobile phase at a flow rate of 1 mL per minute. Mobile phase was firstly purified using a filtration assembly and then sonicated for 10 min at 20 °C in order to avoid gas bubbles. Next, 20 µL samples were injected using a micro-syringe. After 15 min, ultra violet (UV) absorbance was recorded at 254 nm. Sample peaks were analyzed and compared with standard UV absorption data of secondary metabolites at various retention times. Limit of detection (LOD) and limit of quantification (LOQ) of standard aflatoxin were 0.01 ng/mL–100 µg/mL and 0.1 ng/mL–100 µg/mL, respectively.

### 2.7. Statistical Analysis

Mitigation of aflatoxin production and toxin binding capacity of lactobacilli was compared by one-way ANOVA (analysis of variance) followed by Turkey’s multiple comparison test using Graph pad prism 5.0 software (GraphPad Software, San Diego, CA, USA).

## 3. Results

A total of six potential probiotic lactobacilli, including *Lactobacillus gallinarum* PDP 10, *Lactobacillus reuteri* PDP 24, *Lactobacillus fermentum* FYP 38, *Lactobacillus gallinarum* PL 53, *Lactobacillus paracasei* PL 120, and *Lactobacillus gallinarum* PL 149, were procured from the Department of Microbiology, University of Veterinary and Animal Sciences, Lahore, Pakistan. All isolates were Gram-positive rods and catalase negative.

Only three isolates (PL 53, PL 120, and PL 149) had antifungal activity observed by well diffusion assay, as illustrated in Table 1 and Figure 1.

Four isolates (PDP 10, PDP 24, PL 120, and PL 149) showed 100% removal of AFB1, PL 53 caused 55.2% reduction, while FYP 38 showed an enhancing effect on aflatoxin B1 production, as described in Table 2. All isolates showed a varied degree of toxin binding capacities, as described in Table 3 and Figure 2. PL 149 was the most effective binder of aflatoxin B1, with 65% capacity.

## 4. Discussion

Aflatoxins represent a group of fungal secondary metabolites that are of great health and economic importance. In developing countries, greater than five billion people are at risk of chronic exposure to aflatoxins, which are capable of causing liver cancer [4]. Consequently, there is an increasing demand for novel preventive and controlling strategies for aflatoxin contaminations in food and feed. Recent studies have revealed the aflatoxin binding ability of lactobacilli. Many bacteria have been reported as aflatoxin binders, including *Flavobacterium aurantiacum*, *L. plantarum*, *L. pentosus*, and *L. beveris* [20,21,22]. Likewise, *Lactobacillus casei psuedoplantarum* 371, obtained from silage inoculum, inhibited aflatoxin B1 and G1 synthesis by *Aspergillus flavus* subsp*. parasiticus* NRRL 2999 in liquid medium [23]. In a previous study, a mixture of lactobacilli was found to reduce mold growth, germination of spores, and production of aflatoxins by *Aspergillus flavus* subsp. *parasiticus* [24]. A large number of such studies have been reported worldwide, but few related studies have been reported in Pakistan.

The present study can act as a preliminary step in a multistep study to investigate the anti-fungal, anti-aflatoxigenic, and in vitro AFB1 binding capacities of previously characterized indigenous phytase-solubilizing probiotic lactobacilli spp. of poultry and fermented food origin [13] against toxigenic *Aspergillus flavus*. This study identified three probiotic lactobacilli isolates (*Lactobacillus gallinarum* PL 53*, Lactobacillus paracasei* PL 120*,* and *Lactobacillus gallinarum* PL 149) as antifungal agents. Such inhibitory effects may be a result of lactic acid production or physical interaction of lactobacilli with mold. Similar inhibitory effects of *L. acidophilus* ATCC 4495 and *L. brevis* were also demonstrated previously against *Aspergillus flavus* and *Aspergillus parasiticus*, respectively [25,26].

Four isolates (PDP 10, PDP 24, PL 120, PL 149) in the present study ceased aflatoxin production completely, whereas PL 53 showed 55% reduction. On the other hand, FYP 38 showed an enhancing effect on aflatoxin B1 production. These variable results may depict different bacterial cell wall structures. Many other investigators have reported similar results, in which various lactic acid-producing bacteria, including *Lactobacillus*, were capable of inhibiting aflatoxin production, whereas some lactic acid bacteria, like *Lactococcus lactis*, stimulated aflatoxin biosynthesis [27]. Cell wall polysaccharides and peptidoglycans have been considered bacterial tools for mycotoxin binding [28]. Extracellular metabolites of *Lactobacillus casei* KC 324 has been reported to mitigate mold growth and aflatoxin production of *Aspergillus flavus* ATCC 15517 [29]. Commercial silage was once reported to contain inhibitory lactobacilli against aflatoxin B1 and G1 production [30]. *L. plantarum* ATCC 4008*, L. plantarum* 12006, *Lactobacillus plantarum* 299V, *L. paracasei* subsp. *paracasei* LMG 13552, and *L. rhamnosus* VT1 reduced aflatoxin production by 85–92% to 96.3–98.3% [31], whereas in the present study a 100% reduction in AFB1 production by *L. gallinarum* PDP 10 and PL 149, *L. reuteri* PDP 24, and *L. paracasei* PL 120 was observed. It may also be a result of very low aflatoxin production in control conditions as well. Yeast can also act as an effective biocontrol agent against aflatoxins. *S. boulardii* and *S. cerevisiae* individually reduced aflatoxin production by 72.8% and 65.8%, respectively, while their combinations reduced aflatoxin production from 71.1% to 96.1%. Supplementation of peanut grains with combinations of *S. boulardii* plus *L. delbrueckii*, *S. boulardii* and *S. cerevisiae*, *L. delbrueckii* and *S. cerevisiae* showed reduction by 96.1%, 66.7%, and 71.1%, respectively [32]. *Lactobacillus fermentum* PTCC 1744 and *Bifidobacterium bifidum* PTCC 1644 were also reported to reduce aflatoxin production by more than 99% in comparison with controls [6], although this report is contradictory to the present research which revealed the enhancing effect of *Lactobacillus fermentum* on AFB1 production by *A. flavus*.

In the present study, *Lactobacillus gallinarum* PDP 10, *Lactobacillus fermentum* FYP 38, Lactobacillus reuteri PDP 24, *Lactobacillus gallinarum* PL 53, *Lactobacillus paracasei* PL 120, and *Lactobacillus gallinarum* PL 149 showed aflatoxin binding capacities of 51.3%, 56%, 2%, 42%, 28%, and 65%, respectively. These results were quite similar with that of Fazeli et al. [33]. In a previous study, the aflatoxin B1 binding capacities of *Lactobacillus* and *Bifidobacterium* strains were assessed, which were found to range from 5.8% to 31.3% [12]. On the other hand, the present study reported up to 65% AFB1 binding abilities of probiotic lactobacilli. A previous study reported that *Lactobacillus casei* had a 20% AFB1 binding capacity [34], which is less than that of *L. paracasei* PL 120 (28%), whereas *Lactobacillus delbrueckii* subsp. *lactis* was reported to have the maximum antifungal (67.43% reduction) and anti-aflatoxigenic (94.33% reduction) activity against *A. flavus* [35]. Another previous study reported 43.9–64.2% aflatoxin degrading ability of lactobacilli strains [36]. Past investigations revealed similar responses of non-viable and viable cells of *Enterococcus faecium* strains, whose binding abilities were insignificant statistically. Hence, it was hypothesized that AFB1 detoxification of *Enterococcus faecium* is a result of aflatoxin binding to bacterial cell wall; a similar mechanism has been also described by other relevant studies [37]. An in vivo experiment revealed the neutralizing capability of *Lactobacillus casei Shirota* on AFB1 toxicity on the intestine and body weight of host via binding processes [38]. Thus, lactic acid bacteria have been declared as good candidates to prevent aflatoxicosis in farm animals and poultry [9].

## 5. Conclusions

The present study reported the anti-fungal, anti-aflatoxigenic, and AFB1 binding capacity of six indigenously characterized probiotic strains. It was concluded that *L. gallinarum* PL 149 may inhibit the AFB1 production by *A. flavus* and also bind AFB1. *L. gallinarum* PL 149 may be employed for aflatoxin binding in poultry gut after in vivo evaluations.

## Figures and Tables

**Figure 1 animals-09-00166-f001:**
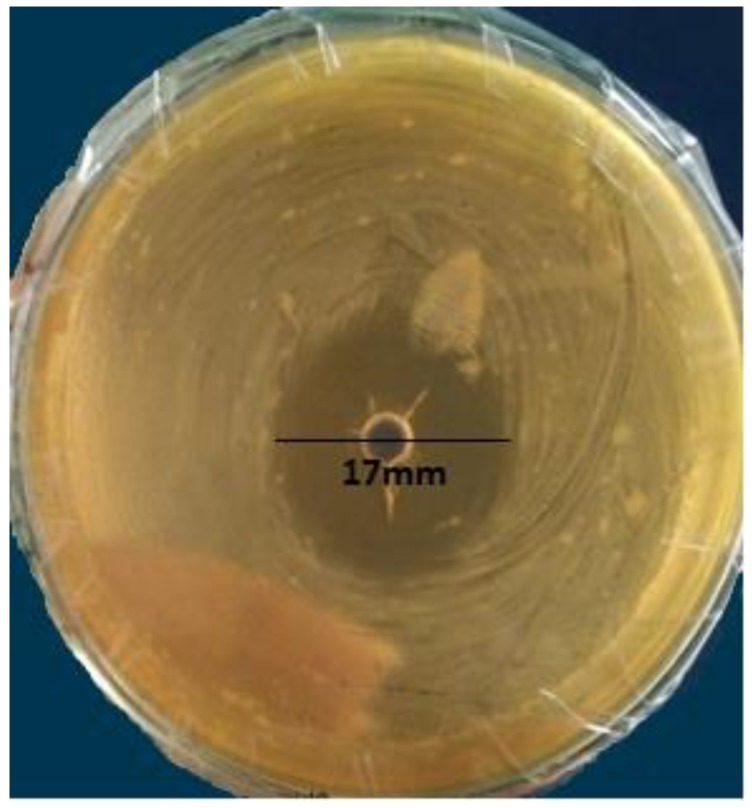
Activity of cell free supernatant of *Lactobacillus gallinarum* PL 149 against *Aspergillus flavus.*

**Figure 2 animals-09-00166-f002:**
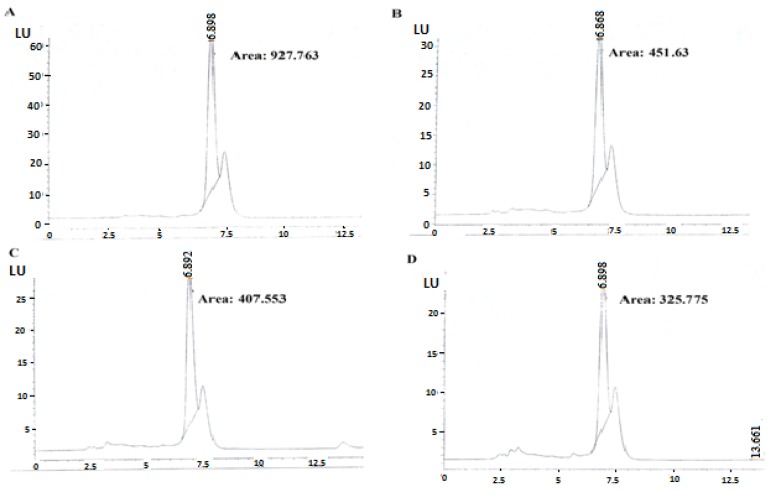
High-performance liquid chromatography chromatograms of aflatoxin B1 present in control and suspension after treatment with lactobacilli: (**a**) Control; (**b**) PDP 10; (**c**) FYP 38; (**d**) PL 149.

**Table 1 animals-09-00166-t001:** Antifungal activity of cell free supernatants of lactobacilli.

Isolates	GenBank Accession #	Zones of Inhibition (mm)	
		pH 4	pH 7
*L. gallinarum* PDP 10	MF980924	NZ	NZ
*L. reuteri* PDP 24	MF980925	NZ	NZ
*L. fermentum* FYP 38	MF980923	NZ	NZ
*L. gallinarum* PL 53	MK182967	13	12
*L. paracasei* PL 120	MK182968	16	14
*L. gallinarum* PL 149	MK182969	17	15

NZ: No zone of inhibition.

**Table 2 animals-09-00166-t002:** Effect of lactobacilli on aflatoxin B1 production.

Isolates	Peak Areas	Quantity of AFB1 (ng/mL)	% Age Reduction
Standard	120.205	100	-
Control	0.58439	0.4	-
*L. gallinarum* PDP 10	ND	ND	100%
*L. fermentum* FYP 38	0.815847	0.6	−39.6%
*L. reuteri* PDP 24	ND	ND	100%
*L. gallinarum* PL 53	0.26124	0.2	55.2%
*L. paracasei* PL 120	ND	ND	100%
*L. gallinarum* PL 149	ND	ND	100%

AFB1: Aflatoxin B1; ND: Not detected.

**Table 3 animals-09-00166-t003:** Aflatoxin B1 binding capacity of probiotic lactobacilli.

Isolates	Peak Areas	Quantity of AFB1 Bound (ng/mL)	% Age Reduction (Binding Capacity)
Standard	108.246	100	-
Control	927.763	857	-
*L. gallinarum* PDP 10	451.63	417.2	51.3%
*L. fermentum* FYP 38	407.553	376.5	56%
*L. reuteri* PDP 24	909.624	840	2%
*L. gallinarum* PL 53	546.523	504.8	42%
*L. paracasei* PL 120	676.472	624.9	28%
*L. gallinarum* PL 149	326.775	301.8	65%

AFB1: Aflatoxin B1.

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
