# Peer review of "Activity and Anti-Aflatoxigenic Effect of Indigenously Characterized Probiotic Lactobacilli against Aspergillus flavus—A Common Poultry Feed Contaminant"

_animals, 2019, doi:10.3390/ani9040166_

Round 1
Reviewer 1 Report
In the present paper I reviewed, the Authors aimed to assess the anti-fungal and anti-aflatoxigenic effect of indigenously characterized probiotic lactobacilli against Aspergillus flavus.
I think that the manuscript fits well the overall scope the journal; moreover, novel and interesting findings have been reported. The results in form of tables and figures have been properly reported and they are significant. The findings were accurately discussed using good and updated references. The language used is also fine.
Based on my opinion, the paper is ready for the acceptance in its present form in Animals.
Author Response
In
the present paper I reviewed, the Authors aimed to assess the
anti-fungal and anti-aflatoxigenic effect of indigenously characterized
probiotic lactobacilli against Aspergillus flavus.
I
think that the manuscript fits well the overall scope the journal;
moreover, novel and interesting findings have been reported. The results
in form of tables and figures have been properly reported and they are
significant. The findings were accurately discussed using good and
updated references. The language used is also fine.
Based on my opinion, the paper is ready for the acceptance in its present form in Animals.
Response: Thank you so much for your kind comments and high admiration of our research.
Reviewer 2 Report
In this manuscript "Anti-fungal and anti-aflatoxigenic effect of indigenously characterized probiotic lactobacilli against Aspergillus flavus” by Nimra Azeem and co-authors presents the results of their studies on effect of six different types of probiotic lactobacilli against Aspergillus flavus and discussed their use in Poultry feed. The present results confirmed that the probiotics are useful in removing toxins from the food and feed.
There are a few points that have to be addressed and rewrite the manuscripts before publishing: Major revision
1. Title: Authors should come up with different title –could include a word “Poultry feed”.
2. Page 2, Line 130: repeated word teratogen teratogenic; authors are suggested to either one of it.
3. Authors are suggested to include few references at line no. 154 in page no. 2, and also recommend to include reference “Manjunath Manubolu et al., 2018: book chapter” entitled “Enzymes as direct decontaminating agents-mycotoxins”-In Enzymes in Human and Animal Nutrition 2018; Pg. 313-330, Academic press as one of the reference. Include proper WHO reference.
4. Include references for methodology sections such as,
Identification of isolates
Antifungal activity of lactobacilli
Aflatoxin B1 extraction.
5. Figure .1 title: sounds like aim/purpose/objective of the work. Need to include proper title.
6. Figure 1 is not clear. Authors are suggested to capture the photographs with clear background without the fingers and finger-prints.
7. I am afraid how authors have calculated the inhibition zone without clear demarcation of the inhibitory zones.
8. Authors should demonstrate about the HPLC method preparation or development and also should clarify the LOD and LOQ of the standard aflatoxin/control samples.
9. Authors are suggested to strengthen the importance and relevance of the present results in the discussion section.
10. Authors should check the conclusion part: the results are really supporting the conclusion? The tested /selected parameters are sufficient for the conclusion?
11. Need to correct/remove the sentence “This research received no funding for APC” from funding source subsection.
12. References should be cited by following journal style/format.
13. Need to check for typographical errors, plagiarism, punctuation, and grammar throughout the manuscript.
Author Response
Response to Reviewer 2 Comments
Point 1: Title: Authors should come up with different title –could include a word “Poultry feed”.
Response 1: We thank the reviewer for this comment and agree to add the word “poultry feed” in title. Title has been modified now as “Activity and anti-aflatoxigenic effect of indigenously characterized probiotic lactobacilli against Aspergillus flavus- a common poultry feed contaminant ”
Point 2: Page 2, Line 130: repeated word teratogen teratogenic; authors are suggested to either one of it.
Response: The word “teratogenic” has been removed from line 130 (page 2)
Point 3: Authors are suggested to include few references at line no. 154 in page no. 2, and also recommend to include reference “Manjunath Manubolu et al., 2018: book chapter” entitled “Enzymes as direct decontaminating agents-mycotoxins”-In Enzymes in Human and Animal Nutrition 2018; Pg. 313-330, Academic press as one of the reference. Include proper WHO reference.
Response: Suggested references have been added.
Point 4: Include references for methodology sections such as,
Identification of isolates
Antifungal activity of lactobacilli
Aflatoxin B1 extraction.
Response: References have been added in the above mentioned sections of Materials and methods as per kind suggestion of reviewers.
Point 5: Figure .1 title: sounds like aim/purpose/objective of the work. Need to include proper title
Response: Title of Figure 1 has been changed as : “Activity of cell free supernatant of Lactobacillus gallinarum PL149 against Aspergillus flavus” as per suggestion.
Point 6: Figure 1 is not clear. Authors are suggested to capture the photographs with clear background without the fingers and finger-prints
Response: We thank the reviewer for this valuable suggestion. A better representative picture had been added in revised manuscript.
Figure 1: Activity of cell free supernatant of Lactobacillus gallinarum PL149 against Aspergillus flavus
Point 7: I am afraid how authors have calculated the inhibition zone without clear demarcation of the inhibitory zones.
Response: We thank reviewer for pointing this discrepancy. We have replaced the figure 1 with a better representative picture and which show clear demarcation of inhibitory zones.
Point 8: Authors should demonstrate about the HPLC method preparation or development and also should clarify the LOD and LOQ of the standard aflatoxin/control samples.
Response: Brief description of HPLC procedure, limit of detection (LOD) and Limit of quantification (LOQ) values has been added in materials and methods section of revised manuscript as per valuable suggestion. Revised manuscript now read as below
“2.6. High Performance Liquid Chromatography (HPLC)
Aflatoxins were quantified by HPLC analysis , as described previously [17]. C18 column was used in in Agilent 1100 Series HPLC Value System. Mixture of acetonitrile, water and methanol was used as mobile phase at a flow rate of 1ml per minute. Mobile phase was firstly purified using a filtration assembly and then sonicated for 10 minutes at 20°C in order to avoid gas bubbles. 20 µl sample was injected using a micro-syringe. After 15 minutes, ultra violet (UV) absorbance was recorded at 254 nm. Sample peaks were analyzed and compared with standard UV absorption data of secondary metabolites at various retention times. Limit of detection (LOD) and limit of quantification (LOQ) of standard aflatoxin were 0.01ng/ml-100µglml and 0.1ng/ml-100µglml, respectively.”
Point 9: Authors are suggested to strengthen the importance and relevance of the present results in the discussion section.
Response: Discussion been improved as per valuable suggestions in revised manuscript.
Point 10: Authors should check the conclusion part: the results are really supporting the conclusion? The tested /selected parameters are sufficient for the conclusion?
Response: We thank the reviewer for this valuable suggestion. We have rationalized and toned down the conclusion in revised manuscript. “Present study reported anti-fungal, anti-aflatoxigenic and AFB1 binding capacity of six indigenously characterized probiotic strains. It is concluded that L. gallinarum PL 149 may inhibit the AFB1 production by A. flavus and also bind AFB1. L. gallinarum PL 149 may be employed for aflatoxin binding in poultry gut after in vivo evaluations ”
Point 11: Need to correct/remove the sentence “This research received no funding for APC” from funding source subsection.
Response: Above mentioned line has been removed from the manuscript, as per kind suggestions
Point 12: References should be cited by following journal style/format.
Response: Reference style has been converted to MDPI style. Journal names has been abbreviated in bibliography.
Point 13: Need to check for typographical errors, plagiarism, punctuation, and grammar throughout the manuscript.
Response: Typographical errors, punctuation and grammar of manuscript have been checked and corrected. Plagiarism has been checked using Turnitin application. There is only 7% plagiarism.
Regards,
Dr. Muhammad Nawaz
Corresponding author
Round 2
Reviewer 2 Report
In this manuscript "Anti-fungal and anti-aflatoxigenic effect of indigenously characterized probiotic lactobacilli against Aspergillus flavus” by Nimra Azeem and co-authors presents the results of their studies on effect of six different types of probiotic lactobacilli against Aspergillus flavus and discussed their use in Poultry feed. The present results confirmed that the probiotics are useful in removing toxins from the food and feed.
Authors have addressed all the questions and I hope now the manuscript look better for publication in the Journal.